# Dysmagnesemia Incidence in Hospitalized Dogs and Cats: A Retrospective Study

**DOI:** 10.3390/ani15081169

**Published:** 2025-04-18

**Authors:** Francesca Perini, Chiara Di Franco, Angela Briganti

**Affiliations:** Department of Veterinary Science, University of Pisa, 56124 Pisa, Italy; francesca.perini@phd.unipi.it (F.P.); angela.briganti@unipi.it (A.B.)

**Keywords:** magnesium, blood gas, veterinary, electrolytes

## Abstract

Few studies have investigated magnesium disorders in veterinary medicine, and most have focused on total magnesium. The aim of the present study was to assess the incidence of dysmagnesemia in hospitalized dogs and cats by focusing on the ionized fraction, which is the biologically active form. Our results showed that hypomagnesemia was common in dogs, even if it was not associated with increased mortality or prolonged hospitalization. In contrast, magnesium disorders were uncommon in cats; however, when hypermagnesemia occurred, the mortality rate increased.

## 1. Introduction

Although in the last few years critical care medicine has given more attention to magnesium, few studies have investigated its concentration in hospitalized veterinary patients [1]. Toll et al. (2002) reported a high incidence of hypomagnesemia in hospitalized cats and found a prolonged hospitalization time in dysmagnesemic patients, together with an increase in mortality [2]. In hospitalized dogs, previous studies reported an incidence of hypomagnesemia of 6.1% [3] and 54% [4], while 13% of dogs were hypermagnesemic [4]. Similarly to feline patients, hypomagnesemia in dogs was associated with a longer period of hospitalization, while hypermagnesemia was related to greater mortality [4]. Recently, Murray et al. (2023) investigated the correlation between total and ionized magnesium in dogs, discovering a prevalence of hypomagnesemia in 67% of cases for ionized magnesium and 18% for total magnesium [5].

In recent years, the scientific community has questioned whether it would be more useful to measure total magnesium or the ionized fraction. Both in humans and dogs, a low correlation between ionized and total magnesium was found in hospitalized patients [5,6,7]. In fact, low total magnesium concentration is often correlated with normal values of ionized magnesium, which could lead to unnecessary supplementation of this electrolyte [7]. Because ionized magnesium is the biologically active form, it is considered the correct fraction to be measured to detect dysmagnesemia in critically ill patients [8]. However, previous studies have mainly focused on total magnesium [2,3,4].

The aim of this study was to retrospectively evaluate the incidence of magnesium disorders in hospitalized dogs and cats by assessing the ionized fraction through venous blood gas analysis. The second goal was to investigate any correlation between dysmagnesemia and the reason for admission, length of hospitalization, and patient’s outcome.

## 2. Materials and Methods

All medical records of dogs and cats admitted to the Veterinary Teaching Hospital of the University of Pisa, from April 2020 to June 2023, were retrospectively reviewed. If the same patient presented to the emergency department multiple times during the study period, only the first visit was considered. Patients were included if they had at least one complete venous blood gas analysis performed at admission (Stat Profile Prime Plus Critical Care Analyzer^®^, Nova Biomedical, Milan, Italy). Patients were excluded if the blood gas analysis was incomplete or if ionized magnesium was not measured.

In addition to magnesium, the following parameters were collected from the venous blood gas analysis performed at admission: pH, sodium, chlorine, calcium, potassium, blood urea nitrogen (BUN), creatinine, lactates, and glycemia.

Patients were classified as hypomagnesemic, normomagnesemic, or hypermagnesemic based on the first blood gas analysis, with normal values for ionized magnesium in the range of 0.5–1 mmol/L for both species according to the machine reference range.

Other recorded data included species, breed, sex, age, weight, reason for hospitalization, presence of comorbidities, outcome (defined as death, either through euthanasia or naturally, or discharge from the ICU), length of hospitalization, and systolic and mean arterial pressure (measured at the time of admission). No limits were imposed on breed, weight, sex, or age.

For statistical purposes, patients were classified into one of the following categories based on the primary cause of admission to the intensive care unit, as follows: cardiovascular, respiratory, gastroenteric, urinary, reproductive, endocrine, hematopoietic, immune-mediated, neurological, musculoskeletal, neoplastic, infectious, and systemic diseases.

### 2.1. Quality Control Procedures

Quality control for the Stat Profile Prime Plus Critical Care Analyzer^®^ (Milan, Italy) was performed according to the manufacturer’s recommendations. Daily calibration was conducted to ensure the accuracy and precision of the device. In addition, control samples with known concentrations of electrolytes and gases were regularly tested to verify the analytical performance of the instrument. If deviations from expected values were detected, corrective actions, such as, for example, recalibration, were taken.

### 2.2. Statistical Analysis

Data distribution was analyzed with a D’Agostino Pearson test; contingency was assessed with a Fisher’s test and a chi-square test. Results were examined with a Mann–Whitney or a Kruskal–Wallis test. A multiple logistic regression was performed to evaluate the outcome in relation to dysmagnesemia.

The chi-square test was used to compare the outcome between dysmagnesemic and normomagnesemic patients. *p* values < 0.05 were considered significant.

## 3. Results

In the study, 430 dogs and 310 cats were enrolled. In dogs, the breeds were mostly mixed breed (*n* = 126), Jack Russell Terrier (*n* = 24), and Golden Retriever (*n* = 23). Furthermore, 191 were females (101 intact, 90 neutered), and 239 were males (201 intact, 38 neutered). The mean age was 7.2 ± 4.5 years, and the median weight was 19 kg (range of 0.5–70 kg).

For dogs, the mean value of magnesium was 0.54 ± 0.12 mmol/L. Upon admission, 153 dogs were hypomagnesemic (35.5%), five were hypermagnesemic (1.1%), and 272 were normomagnesemic (62.2%). No differences (*p* = 0.9) were found in the chi-square test between survivors and non-survivors for magnesium concentration, while males showed a significantly higher (*p* = 0.0006) incidence of hypomagnesemia than females.

In dogs, hypomagnesemia showed a higher incidence in neurological (51%), neoplastic (50%), and endocrine (42%) pathologies, but no significant differences were found. Moreover, other blood gas parameters were analyzed for statistical significance because of their known interactions with magnesium. In fact, pH (*p* = 0.00073), Na^+^ (*p* = 0.0002), Ca^2+^ (*p* = 0.041), and K^+^ (*p* < 0.0001) were significantly lower in hypomagnesemic dogs.

Furthermore, multiple logistic regression in dogs showed that dysmagnesemia in dogs was not associated with lower survival, a longer period of hospitalization, or changes in systolic arterial pressure (SAP) or mean arterial pressure (MAP).

Moreover, 57.79% of hypomagnesemic dogs had concurrent hypertension, 20.13% had concurrent hypocalcemia, and 33.77% had concurrent hyponatremia.

Among hypomagnesemic dogs with concurrent hypertension, 24.72% belonged to the gastroenteric category, as did 35.48% of hypomagnesemic dogs with concurrent hypocalcemia and 36.54% of hypomagnesemic dogs with concurrent hyponatremia.

Regarding cats, the most represented breeds were European shorthair (*n* = 269), Persian cats (*n* = 12), and Maine Coon (*n* = 6). Furthermore, 114 were females (30 intact, 84 neutered), and 196 were males (47 intact, 149 neutered). The mean age was 6.8 ± 4.9 years, and the median weight was 4.5 kg (range of 0.3–10 kg).

In cats, the mean value of magnesium was 0.69 ± 0.19 mmol/L. Upon admission, 21 cats were hypomagnesemic (6.8%), 25 were hypermagnesemic (8%), and 264 were normomagnesemic (85.2%). Hypermagnesemic cats had a mortality rate 2.3 times higher (*p* = 0.0001) than that of normomagnesemic and hypomagnesemic cats.

In cats, hypermagnesemia had a higher incidence in endocrine (28.6%), systemic (13.6%), and urinary (12.9%) pathologies, but no significant differences were highlighted.

Moreover, hypermagnesemic cats had significant increases in BUN (*p* < 0.0001), creatinine (*p* = 0.0004), and K^+^ (*p* = 0.0005), along with significant decreases in pH (*p* < 0.0001) and Ca^2+^ (*p* = 0.0007).

Multiple logistic regression in cats showed that dysmagnesemia was associated with a lower rate of discharge (dysmagnesemic cats had an odds ratio of 7.3 for not being discharged), but it was not associated with a longer period of hospitalization or changes in SAP or MAP.

Moreover, 64% of hypermagnesemic cats had concurrent hypertension, 20% had concurrent hypocalcemia, and 52% had concurrent hyponatremia.

Among hypermagnesemic cats with concurrent hypertension, 56.25% belonged to the urinary category. Similarly, 40% of those with concurrent hypocalcemia also fell into the urinary category. Among hypermagnesemic cats with concurrent hyponatremia, 23.08% belonged to the gastroenteric category, and 23.08% belonged to the urinary category.

Figure 1 and Figure 2 show the incidence of hypomagnesemia for dogs and hypermagnesemia for cats for each category. However, detailed definitive diagnoses within each category were not available due to limitations in the medical records.

In Table 1 and Table 2, the mean values of ionized magnesium and related parameters are presented for hypomagnesemic dogs and hypermagnesemic cats, respectively.

## 4. Discussion

This is the first study regarding the incidence of dysmagnesemia in critically ill dogs and cats that evaluates ionized magnesium instead of total magnesium. The increasing availability of blood gas analyzers in recent years has made assessing the ionized fraction of this cation more accessible in clinical practice.

Understanding disease predisposition to dysmagnesemia is essential, as alterations in magnesium concentrations could be associated with longer hospital stays or increased mortality [2,3,4].

In our study, upon admission, 153 dogs were hypomagnesemic (35.5%), five were hypermagnesemic (1.1%), and 272 were normomagnesemic (62.2%). Murray et al. (2023) reported an incidence of 67% for hypomagnesemia and 4% for hypermagnesemia when evaluating ionized magnesium in 45 hospitalized dogs [5]. The difference in dysmagnesemia incidence could be due to the number of subjects involved and the different cutoff values used. However, even in our study, dogs exhibited a higher incidence of hypomagnesemia than hypermagnesemia.

In this study, hypomagnesemia was not associated with increased mortality, in contrast with findings in previous studies [4]. In human medicine, hypomagnesemia is common in hospitalized patients [9,10,11,12,13] and associated with poor outcomes [14,15]. Moreover, hypomagnesemia in human medicine has a higher incidence in males [10,16], as observed in our study.

Hypomagnesemic dogs showed significantly lower concentrations of Na^+^, Ca^2+^, and K^+^, according to previous studies [3]. Magnesium acts as a cofactor for more than 300 enzymes [17], and it is fundamental to the function of the Na^+^/K^+^ ATPase pump. If hypomagnesemia occurs, pump activity decreases, causing sodium to remain in the intracellular space while potassium moves from the intracellular to the extracellular fluid, which is then excreted in the urine [18]. In fact, in the kidneys, hypomagnesemia increases the activity of ROMK (Renal Outer Medullary Potassium) channels, resulting in elevated potassium excretion in the tubules [18,19,20].

Furthermore, hypomagnesemia is associated with higher aldosterone levels [21], which stimulates urinary potassium excretion [22].

In fact, potassium deficiency is often unresponsive to supplementation until hypomagnesemia is corrected [23].

Regarding calcium, magnesium is a cofactor for cAMP production in the parathyroid glands. When hypomagnesemia occurs, cAMP production declines, leading to reduced parathyroid hormone (PTH) secretion and, consequently, lower blood calcium concentrations [24,25,26]. Moreover, reduced cAMP production may lead to organ resistance to PTH [27].

Both calcium and magnesium are divalent cations, so they share similar renal reabsorption pathways. Specifically, the Calcium-Sensing Receptors (CaSR) in the kidneys detect their concentrations and adjust renal absorption accordingly [28,29].

Our study revealed a higher incidence of hypomagnesemia in dogs with neurological (51%) disorders, followed by neoplastic (50%), and endocrinological (42%) conditions. In our veterinary hospital, the most common neurological conditions leading to hospitalization include seizures, vestibular syndrome, disc herniations, and traumatic brain injury. In the nervous system, magnesium is very important for neuronal excitability and neuromuscular transmission, and it is necessary to protect nervous cells against excessive excitation, which can lead to cellular death [30,31]. This cation is essential for the N-methyl-D-aspartate (NMDA) receptor function, as magnesium blocks the receptor and prevents ion influx. When depolarization occurs and glutamate interacts with the NMDA receptor, magnesium is displaced, allowing ions, mostly calcium, to enter [30]. When hypomagnesemia occurs, glutamatergic neurotransmission increases; in fact, low magnesium concentration is correlated with higher neuronal excitability and neuromuscular conduction, leading to muscle weakness, spasms, seizures, and tetany [1,32,33]. In human medicine, altered glutamatergic neurotransmission and hypomagnesemia are linked to various neurological conditions, including migraine, epilepsy, Alzheimer’s, Parkinson’s disease, stroke, depression, and anxiety [34]. In the secondary phase of traumatic brain injury (TBI), a massive release of glutamate occurs with overstimulation of NMDA receptors. Consequently, a great influx of calcium occurs, leading to magnesium depletion and cellular death [35,36]. Therefore, hypomagnesemia is often observed in TBI and correlates with poor neurological outcomes [37,38].

In some neurological disorders, magnesium supplementation has shown a neuroprotective effect, leading to better outcomes. In particular, oral magnesium supplementation may reduce seizures in epileptic children [39,40,41], while in TBI, magnesium supplementation results are mixed, as some studies have reported worse outcomes [42], while others have documented improvements [43].

In our study, hypomagnesemia had 50% incidence in neoplastic diseases. This could be attributed to several causes. First, magnesium is crucial for cellular metabolism [44,45]; thus, when hypomagnesemia occurs, cellular proliferation is inhibited [46]. However, neoplastic cells do not appear to be inhibited by low magnesium concentrations; instead, they tend to accumulate this cation [47], likely due to their overexpression of TRPM7 (Transient Receptor Potential Melastatin), a channel involved in active transcellular magnesium absorption [48]. The great accumulation of magnesium provides an advantage for neoplastic cells, as the cation is involved in many cellular metabolic pathways [49].

Moreover, hypomagnesemia may contribute to the development of neoplastic diseases by increasing inflammation and oxidative stress, which lead to a higher concentration of free radicals and, consequently, DNA damage and inefficient DNA repair mechanisms. This genetic instability can contribute to the development of neoplasia [50,51].

Additionally, hypomagnesemia is a common side effect of certain chemotherapeutic agents, and it may exacerbate other side effects of these drugs. For example, cisplatin is associated with hypomagnesemia due to its nephrotoxicity, which increases magnesium waste [52]. Because of its important side effects for the kidneys, cisplatin is no longer used, as it has been replaced by other drugs, as carboplatin, with fewer renal side effects. However, even carboplatin seems to be associated with hypomagnesemia [53].

Furthermore, during thalidomide therapy, hypomagnesemia may worsen myocardial toxicity, probably through loss of endogenous antioxidants or increased thalidomide toxicity [54].

In our study, endocrinological disorders showed a 42% incidence of hypomagnesemia. Magnesium is important for some hormones’ secretion, and, at the same time, its homeostasis is influenced by other hormones. Aldosterone, a mineralocorticoid hormone produced by the adrenal cortex, regulates water and sodium reabsorption. Hyperaldosteronism can be due to adrenal hyperplasia or neoplasia, or it can attributed to the activation of the renin–angiotensin system because of cardiovascular or renal failure [55,56]. It seems that aldosterone also increases magnesium clearance and excretion in urine [57], so hyperaldosteronism may cause hypomagnesemia [1]. In fact, one study reported that rats with induced hyperaldosteronism had hypomagnesemia due to increased urinary and fecal magnesium waste [58].

In human medicine, types II diabetes is often related to hypomagnesemia. In particular, even if magnesium absorption is maintained, urinary excretion of this electrolyte seems to be increased [59,60]. On the other hand, low magnesium concentration may impair insulin secretion from pancreatic beta cells, as magnesium acts as a cofactor [61], and it may increase tissue insulin resistance [59]. However, Fincham et al. (2004) [62] did not find significant hypomagnesemia in diabetic dogs in their study. Nevertheless, they analyzed ionized magnesium only at the time of admission and not during hospitalization, so they assumed that the magnesium concentration could have changed during the hospital stay [62].

As previously mentioned, magnesium serves as a cofactor for cAMP production in the parathyroid glands, so reduced magnesium concentrations may lead to decreased PTH production and organ resistance to PTH [24,25,26,27]. Hypoparathyroidism, along with hypocalcemia, may also contribute to hypomagnesemia, as PTH increases magnesium absorption in the kidneys and in the gut and its release from the bones [63,64].

Although hypomagnesemia did not have a high incidence in the gastroenteric category, the results showed that 24.72% of those patients were hypertensive, 35.48% were hypocalcemic, and 36.54% were hyponatremic. The primary symptom of gastroenteric disorders is diarrhea, which can lead to the loss of electrolytes and fluids. Additionally, bowel inflammation can impair mineral absorption, contributing to hypomagnesemia, hypocalcemia, and hyponatremia [1].

Furthermore, dehydration and abdominal pain caused by diarrhea can lead to increased sympathetic nervous system activity and activation of the renin–angiotensin–aldosterone system. This results in vasoconstriction and an increase in plasma volume. Inflammation itself can also induce vasoconstriction due to oxidative stress [65,66,67].

Concerning cats, although hypermagnesemia did not have a high incidence (8%), patients with this disorder had a poorer outcome. In fact, their mortality rate was 2.3 times higher than that of the other two categories. We found a lower incidence of hypermagnesemia than Toll et al. (2002), who reported a prevalence of 18% in hospitalized cats [2]. This difference may be explained by the fact that we considered a larger sample, specifically, 310 cats, whereas they considered only 57 patients.

Another study reported an incidence of hypermagnesemia in cats with chronic kidney disease of 6% [68], which is quite similar to our results.

In our study, hypermagnesemia was more common in endocrine disorders (28.6%). This result differs from findings documented in the literature. Hospitalized cats with endocrine disorders typically suffer from diabetes mellitus (DM), diabetic ketoacidosis (DKA), and hyperthyroidism. In cats with DM and DKA, magnesium deficiency is the most common dysmagnesemia, possibly due to an increased urinary fractional excretion of magnesium [9].

In human medicine, diabetic patients also frequently suffer from hypomagnesemia, although hypermagnesemia in diabetic patients has been associated with a higher incidence of microvascular complications [69].

It is possible that cats with DKA and metabolic acidosis develop hypermagnesemia because of the shift from the intracellular to the extracellular space [70]. This shift occurs because during metabolic acidosis, hydrogen ions’ (H^+^) concentration in the blood increases. To maintain acid–base balance, H^+^ ions are exchanged with intracellular cations, such as potassium and magnesium, displacing them from cells and leading to hypermagnesemia [71].

Hyperthyroidism can also enhance urinary excretion of magnesium, leading to hypomagnesemia [72]. On one hand, renal blood flow increases due to the positive chronotropic effect [73], the inotropic effect [74], and a decrease in systemic vascular resistance [75] induced by thyroid hormones. Additionally, these lead to greater production of nitric oxide in the kidneys, causing intrarenal vasodilatation and thus enhancing renal blood flow [76].

On the other hand, the glomerular filtration rate increases [77] due to increased renal blood flow, as well as the stimulatory effects of thyroid hormones on renin–angiotensin–aldosterone system (RAAS) activation [78,79].

It is possible that polyphagia, a common sign in hyperthyroid cats, increases oral intake of magnesium, which could compensate for urinary losses [80].

Moreover, chronic kidney disease and hyperthyroidism are often comorbid conditions [81]. In end-stage chronic kidney disease, hypermagnesemia may occur due to a reduction in GFR [82].

Sepsis was the most common finding in the systemic category. Mortality rate in septic cats is reported to be 40% [83]. In our study, this category had a high incidence of hypermagnesemia (13.6%). In children with sepsis, magnesium excess is associated with increased mortality [84]. First, cardiovascular effects must be considered, as magnesium acts as a calcium antagonist, blocking L-type calcium channels in the smooth muscle cells of blood vessels. Because calcium cannot enter the cells and is essential for muscle contraction, its absence reduces contractions in vascular smooth muscle cells, leading to vasodilation [85,86,87].

Moreover, magnesium excess can lead to arrhythmias, as this cation influences calcium and potassium channels, as well as the Na^+^/K^+^ ATPase pumps of myocardial cells [33,88].

Furthermore, hypermagnesemia can lead to muscle weakness and flaccid paralysis, which can also affect respiratory muscles, leading to hypoventilation and life-threatening respiratory depression [70,89,90,91].

In summary, the effects of hypermagnesemia may contribute to hemodynamic and systemic instability in septic patients, resulting in an increased risk of mortality.

We found that magnesium excess also had a high incidence in urinary disorders (12.9%), and it was associated with increased BUN, creatinine, and K^+^, while it was correlated with a significant decrease in pH and Ca^2+^. Although the association between hyponatremia and hypomagnesemia was not statistically significant, 23.08% of cats with both disorders belonged to the urinary category, and another 23.08% belonged to the gastroenteric category. This may be explained by the fact that hyponatremia can result from sodium losses through both the kidneys and the gastrointestinal tract [92].

As previously mentioned, end-stage chronic kidney disease is associated with hypermagnesemia due to the loss of renal function. In this situation, the kidneys cannot efficiently excrete magnesium in urine, leading to an accumulation of this electrolyte in the blood [82].

Urethral obstruction is a medical emergency in which cats cannot urinate because of the presence of urethral plugs, uroliths, or idiopathic factors [93]. In this condition, electrolyte disorders, such as hyperkalemia, hyperphosphatemia, hypocalcemia, and hypermagnesemia, may develop [93,94,95].

The resultant metabolic acidosis frequently observed in obstructed cats may exacerbate hypermagnesemia for the reasons previously mentioned [70,71]. Unlike chronic metabolic acidosis, which leads to renal magnesium wasting due to the reduced expression of TRPM6 receptors in the kidneys [96], acute metabolic acidosis enhances magnesium absorption in the renal tubules, as magnesium is fundamental to compensate for H+ ions’ concentration. Further studies are needed, but it is possible that TRPM6 receptors are not rapidly downregulated.

The significant increase we found in hypermagnesemic cats for BUN, creatinine, and K^+^ can be explained by the high incidence of this electrolyte disorder in renal and urinary disorders [93,97]. Furthermore, renal dysfunction is common in septic cats [83], hyperthyroidism [81], and DKA, in which azotemia may result from dehydration or pre-existing renal insufficiency [98].

The decrease in pH and Ca^2+^ can also be attributed to the high prevalence of renal and urinary disorders [93]. Additionally, acidosis and hypocalcemia are frequently observed in septic patients. Hypocalcemia in sepsis may result from reduced conversion of vitamin D to calcitriol, its active form, in the kidneys, leading to decreased intestinal calcium absorption, reduced renal reabsorption, and, consequently, lower blood calcium concentrations [99]. Finally, hypocalcemia may also occur in DKA due to increased urinary excretion and metabolic acidosis, which leads to greater calcium binding to albumin [100].

Lastly, hypermagnesemia in cats was not associated with longer period of hospitalization or changes in blood pressure. Toll et al. (2002) found a longer period of hospitalization in both hyper- and hypomagnesemic cats [2], which is in contrast to the results of our study. This difference may be due to the smaller number of patients Toll et al. considered in their study, as they enrolled 57 cats admitted to the intensive care unit, while we analyzed the medical records of 310 feline patients.

Regarding blood pressure, given the high incidence of hypermagnesemia in urinary and endocrine categories, hypertension might be expected. In fact, high blood pressure frequently occurs in kidney disease [101,102,103,104] and hyperthyroidism [101,105]. Conversely, patients with septic shock suffer from hypotension [106,107]. The contrasting tendencies observed among different categories may explain the lack of statistically significant differences in blood pressure, despite the fact that 56.25% of hypermagnesemic cats with concurrent hypertension belonged to the urinary category. Further studies are needed to evaluate the presence of significant differences within each category.

In our study, hypomagnesemia in dogs did not appear to influence the prognosis, as it was not associated with increased mortality. However, in cats, hypermagnesemia was linked to higher mortality rates. This finding suggests that elevated magnesium concentrations may be indicative of a more severe clinical condition. Additionally, we observed that cats with hypermagnesemia also had a lower pH, which may contribute to the increased mortality observed. In fact, both hypermagnesemia and acidosis can negatively affect cardiovascular and respiratory function [1,70], potentially exacerbating the clinical outcome. Therefore, the association between hypermagnesemia and acidosis in cats may be a marker for poor prognosis and increased mortality.

Interestingly, the incidence of dysmagnesemia in both dogs and cats would have been different if reference ranges reported in the literature had been applied instead of those provided by the blood gas analyzer. In particular, considering a reference range for ionized magnesium of 0.43–0.6 mmol/L for dogs and 0.43–0.7 mmol/L for cats [108], hypomagnesemia would have been observed in 12% of dogs and in 2.6% of cats, while hypermagnesemia would have been observed in 22.8% of dogs and 30.6% of cats. However, we followed the reference range provided by the machine (0.5–1 mmol/L), as typically recommended, to ensure consistency with the analyzer’s internal calibration and clinical use. This highlights the importance of correctly establishing reference ranges, as their variability may significantly impact the diagnosis and prevalence of dysmagnesemia. In this study, the use of the machine reference range was considered the most appropriate approach, as it aligned with the manufacturer’s calibration standards.

This study has several limitations. First, being conducted at a single center may affect its generalizability. Moreover, evaluating the presence of significant differences in each category for hospitalization length, outcomes, electrolytes, and other parameter disorders would be of interest. It would also be valuable to analyze the incidence of dysmagnesemia for each pathology within the categories.

Furthermore, the retrospective nature of the study may have resulted in incomplete or inconsistent documentation of diagnoses and patient management. This could have affected the accuracy of our data, particularly regarding the assessment of magnesium concentrations and the categorization of underlying diseases. Additionally, the dependence on medical records for diagnosis documentation may create bias, as diagnoses were not made with the aim of specifically addressing the study’s objectives.

Moreover, we considered all patients hospitalized in the intensive care unit as critically ill, without using more specific criteria for inclusion. This broad definition could have affected the interpretation of the results by not fully capturing the severity of illness in each patient.

In addition, the incidence of hypomagnesemia may have been overestimated due to the administration of therapies that can reduce electrolyte concentration, such as proton pump inhibitors, diuretics, and chemotherapeutic agents. A prospective study would be necessary to evaluate the impact of such therapies and obtain a more accurate estimate of dysmagnesemia.

Finally, another limitation of this study is that renal and urinary conditions, including urethral obstruction, Acute Kidney Injury (AKI), chronic kidney disease (CKD), and End-Stage Renal Disease (ESRD), were grouped together under the urinary category for simplicity. These conditions may influence ionized magnesium concentrations differently, and their inclusion as a single category could potentially obscure their individual effects. Further studies that differentiate these conditions would provide more detailed insights into their specific impact on ionized magnesium levels.

## 5. Conclusions

Dysmagnesemia was not associated with prolonged hospitalization in either dogs or cats. Moreover, dogs presented a higher incidence of hypomagnesemia, which was not associated with increased mortality. In contrast, in cats, although hypermagnesemia had a low incidence (8%), it was associated with increased mortality.

## Figures and Tables

**Figure 1 animals-15-01169-f001:**
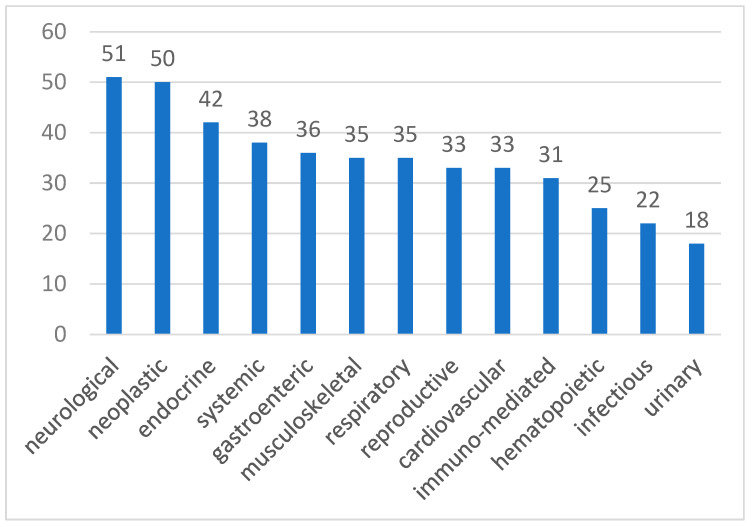
Incidence of hypomagnesemia (%) in dogs.

**Figure 2 animals-15-01169-f002:**
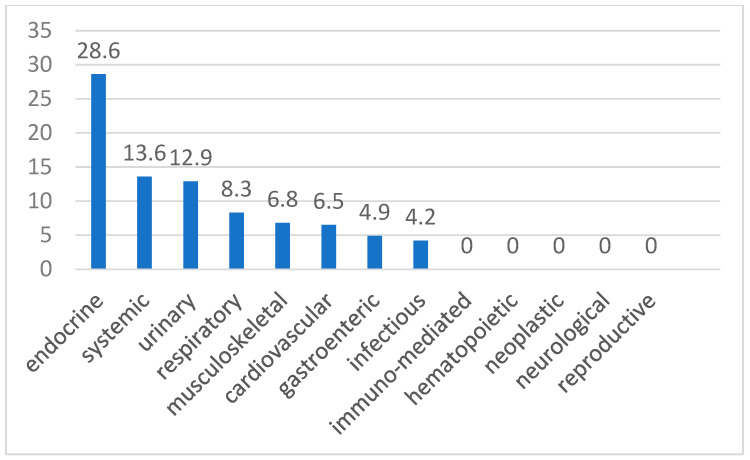
Incidence of hypermagnesemia (%) in cats.

**Table 1 animals-15-01169-t001:** Mean values of analytes and parameters related to ionized magnesium in hypomagnesemic dogs.

Parameter	Mean Value
Mg^2+^	0.44 ± 0.05 mmol/L
pH	7.39 ± 0.06
Na^+^	141.72 ± 6.4 mmol/L
Ca^2+^	1.27 ± 0.12 mmol/L
K^+^	3.99 ± 0.54 mmol/L
BUN	18.15 ± 9.79 mg/dL
Creatinine	1.39 ± 1.58 mg/dL
SAP	149 ± 22 mmHg
MAP	113 ± 17 mmHg

**Table 2 animals-15-01169-t002:** Mean values of analytes and parameters related to ionized magnesium in hypermagnesemic cats.

Parameter	Mean Value
Mg^2+^	1.2 ± 0.5 mmol/L
pH	7.18 ± 0.22
Na^+^	147.17 ± 9.27 mmol/L
Ca^2+^	1.19 ± 1.44 mmol/L
K^+^	4.85 ± 2 mmol/L
BUN	89.52 ± 79.7 mg/dL
Creatinine	3.54 ± 3.52 mg/dL
SAP	160 ± 31 mmHg
MAP	123 ± 30 mmHg

## Data Availability

The data presented in this study are available in this article.

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
