# Peer review of "Dysmagnesemia Incidence in Hospitalized Dogs and Cats: A Retrospective Study"

_animals, 2025, doi:10.3390/ani15081169_

Round 1

Reviewer 1 Report

Comments and Suggestions for Authors

The work lends itself to confusion since it touches on two topics, of which two articles could have been developed, the first is the magnesium reference rate and the second is hypo and hypermagnesemia. I consider it important to place greater emphasis on the separation of these topics in order to make it easier for the reader.

Author Response

  1. The work lends itself to confusion since it touches on two topics, of which two articles could have been developed, the first is the magnesium reference rate and the second is hypo and hypermagnesemia. I consider it important to place greater emphasis on the separation of these topics in order to make it easier for the reader.

Dear reviewer,

Thank you for your suggestion. We have decided to follow your advice and create a separate article on magnesium reference range to improve clarity and understanding.

Reviewer 2 Report

Comments and Suggestions for Authors

Dear authors,
Thank you for submitting your study on magnesium disorders in critically ill small animals patients. I found the topic interesting and unusual. However I have a few comments and suggestions before it can be accepted as a paper in my eyes. 

M&M:

  • What was the reason for anesthesia of healthy cats? Please describe.
  • How did you decide the patient was « critically ill »? Please report the detailled criteria for inclusion.
  • Can you state how you determined the number of healthy cats that were needed to determine the reference range of ionized magenisum? Which statistical test was used? how did you make sure those cats are not predisposed to hypomagnesemia for instance
  • Is the reference range from you blood gas analyser from human medicine? If yes please note it.

Results:

  • Incidence of dysmagnesemia upon different disorders: I am missing the p values of your test. What the incidence significantly higher in those diseases ?
  • I think it would be interesting to look at your cat results in the light of the reference range that you calculated: if you apply your « new » reference range, do you results change? You could add a paragraph in which you look at your new reference range. 

Discussion:

  • Mg and endocrine disease in dogs. It would be interesting to know if the diseases that your patients had were in any reason related to aldosterone secretion for instance. Could you retrieve the most common endocrine diseases you patients had?
  • I think globally the discussion is too long. I would reduce all the details from humans medicine and the heavy pathophysiology and focus more on the influence of Mg on prognosis. 
  • Please discuss the possible association of pH and Mg and the mortality in cats.
  • The limitations are not limitations. Please comment on your real limitations: retrospective nature and documentation of diagnosis for instance, etc...
    I suggest the limitations that you state could be added to the results.

I hope this suggestion will help you improve your manuscript and I am looking forward reading the new version.
Kind regards

Comments on the Quality of English Language

29-30: correct grammar, sentence not clear

42: date of the citation is missing

Author Response

Dear authors,

Thank you for submitting your study on magnesium disorders in critically ill small animals patients. I found the topic interesting and unusual. However I have a few comments and suggestions before it can be accepted as a paper in my eyes. 

M&M:

  1. What was the reason for anesthesia of healthy cats? Please describe.

Dear reviewer, thank you for your feedback. We decided to analyze ionized magnesium in anesthetized healthy cats because, in our hospital, blood samples for blood gas analysis are typically collected after premedication to minimize stress in our patients. Since this is a retrospective study, it was not possible to analyze the electrolyte concentration in awake patients.

Anyway, we have decided to remove the part regarding the magnesium reference range in healthy cats in order to create a separate article, as suggested by one of the reviewers.

  1. How did you decide the patient was « critically ill »? Please report the detailled criteria for inclusion.

 In our study, we considered patients to be critically ill if they were hospitalized in the intensive care unit. However, we now realize that a more detailed set of criteria for inclusion could have provided a clearer definition of critically ill patients. We revised the manuscript to add this as a limitation of the study. Lines 391- 417.

  1. Can you state how you determined the number of healthy cats that were needed to determine the reference range of ionized magenisum? Which statistical test was used? how did you make sure those cats are not predisposed to hypomagnesemia for instance

Since we have decided to remove the section regarding the magnesium reference range from this article to create a separate publication, we are not addressing the determination of the number of healthy cats or the statistical methods used for reference range calculation in this study. However, we ensured that all included cats were healthy based on clinical examination and blood tests and were not receiving any treatment.

  1. Is the reference range from you blood gas analyser from human medicine? If yes please note it.

The reference range for ionized magnesium used in the study of dysmagnesemia is that of the blood gas analyzer which is specific for dogs and cats. The machine is specific for veterinary patients.

Results:

  1. Incidence of dysmagnesemia upon different disorders: I am missing the p values of your test. What the incidence significantly higher in those diseases ?

We are sorry, the difference was not significant , we corrected the sentence in the text. Line 111.

  1. I think it would be interesting to look at your cat results in the light of the reference range that you calculated: if you apply your « new » reference range, do you results change? You could add a paragraph in which you look at your new reference range. 

Thank you for your suggestion. As recommended by one of the reviewers, we have removed the section regarding the magnesium reference range in healthy cats from this article to create a separate publication on this topic. For this reason, we did not apply the calculated reference range to our results in the current study. However, we appreciate your comment, and we will take this aspect into consideration in our future work.

Discussion:

  1. Mg and endocrine disease in dogs. It would be interesting to know if the diseases that your patients had were in any reason related to aldosterone secretion for instance. Could you retrieve the most common endocrine diseases you patients had?

Unfortunately, it is not possible to retrieve the incidence of the various endocrinopathies within the category of endocrine disorders. A prospective study may be necessary to evaluate the relationship with aldosterone secretion.

  1. I think globally the discussion is too long. I would reduce all the details from humans medicine and the heavy pathophysiology and focus more on the influence of Mg on prognosis. 

Thank you for your feedback. While we understand your point regarding the length of the discussion, we believe that including details from human medicine and pathophysiology is important for providing a broader context and a deeper understanding of the mechanisms behind dysmagnesemia. This information helps the reader to better appreciate the relevance of the findings in veterinary medicine. Regarding the influence of magnesium on prognosis, we acknowledge that further discussion could be beneficial, and we can address this in more detail as needed.

  1. Please discuss the possible association of pH and Mg and the mortality in cats.

Thank you for your suggestion. A discussion on the possible association between acidosis, magnesium, and mortality will be included to enhance clarity. Lines 383-390.

  1. The limitations are not limitations. Please comment on your real limitations: retrospective nature and documentation of diagnosis for instance, etc...

I suggest the limitations that you state could be added to the results. 

Thank you for your comment. We acknowledge that the retrospective nature of the study and the documentation of diagnosis are important limitations, and we have now included them in the manuscript as suggested. However, we believe that the other aspects we previously discussed also represent relevant limitations of our study. For this reason, we prefer to keep them in the limitations section rather than move them to the results, as they provide a comprehensive overview of the study’s limitations.

Comments on the Quality of English Language

  1. 29-30: correct grammar, sentence not clear. Corrected, thank you. 
  2. 42: date of the citation is missing Corrected, thank you.
  1. Why not to put a graph of hypermagnesemia in dogs and hypomagnesemia in cats to compare between the species?

The graph for hypermagnesemia in dogs was not included because the incidence of this disorder was low. In contrast, in cats, although the incidence of hypomagnesemia was slightly lower than that of hypermagnesemia, we did not create a graph for it, as hypermagnesemia had a significant impact on mortality. Moreover, in our view, a direct comparison between the two species may not be meaningful, as the disorders and their clinical implications differ between dogs and cats.

  1. Which type of neoplasia in hypomagnesemic dogs?

Given the retrospective nature of the study, it was not possible to accurately classify the specific types of neoplasia in hypomagnesemic dogs. However, the neoplasia category includes a range of different tumor types.

Reviewer 3 Report

Comments and Suggestions for Authors

Dear Authors,

This manuscript is well-written and effectively addresses gaps in the existing literature, highlighting the importance of ionized magnesium. The authors have done a phenomenal job addressing the multifaceted aspects of dysmagnesemia. However, I have several comments that could further enhance the manuscript and the study design.

A significant portion of the study and its conclusions rely heavily on the reference interval of ionized magnesium (iMg) in cats, which is established within the study. Ideally, reference intervals should be established based on ASVCP guidelines. However, the reference interval used in this study is presented based on the 2.5th and 97.5th percentiles. I recommend that the authors re-establish the reference interval of iMg with a confidence interval for both the lower and upper ends of the range, in accordance with ASVCP guidelines.

Both ionized and total magnesium levels are influenced by multiple factors, including medications such as diuretics, corticosteroids, proton pump inhibitors, and chemotherapeutic agents. The manuscript does not specify whether the dogs and cats with hypomagnesemia received enteral or parenteral magnesium supplementation. Did the authors address these confounding factors when investigating the association between dysmagnesemia and mortality? If not, it seems reasonable to mention this as a limitation.

The authors noted that the iMg levels at admission and during hospitalization were collected from medical records. Could the authors please clarify if they made a distinction between admission and hospitalization when assessing the outcomes related to iMg?

Regarding statistical analysis: Could the authors clarify how the association between dysmagnesemia and outcomes was assessed? For example, would regression analysis be appropriate for assessing the likelihood of mortality in relation to the degree of dysmagnesemia and other outcomes?

Please ensure that the Materials and Methods section of the manuscript clarifies whether a range of renal/urinary diseases (e.g., urethral obstruction, ACKD, AKI, CKD, ESRD) were included under the "urinary" category. It could be confusing for readers if all of these conditions, which impact iMg differently, were grouped into this category.

Author Response

This manuscript is well-written and effectively addresses gaps in the existing literature, highlighting the importance of ionized magnesium. The authors have done a phenomenal job addressing the multifaceted aspects of dysmagnesemia. However, I have several comments that could further enhance the manuscript and the study design.

  1. A significant portion of the study and its conclusions rely heavily on the reference interval of ionized magnesium (iMg) in cats, which is established within the study. Ideally, reference intervals should be established based on ASVCP guidelines. However, the reference interval used in this study is presented based on the 2.5th and 97.5th percentiles. I recommend that the authors re-establish the reference interval of iMg with a confidence interval for both the lower and upper ends of the range, in accordance with ASVCP guidelines.

Dear reviewer, thank you for your feedback. We acknowledge the importance of establishing reference intervals following ASVCP guidelines. However, based on the suggestion of another reviewer, we have decided to remove the reference range section from this article and address it in a separate publication.

  1. Both ionized and total magnesium levels are influenced by multiple factors, including medications such as diuretics, corticosteroids, proton pump inhibitors, and chemotherapeutic agents. The manuscript does not specify whether the dogs and cats with hypomagnesemia received enteral or parenteral magnesium supplementation. Did the authors address these confounding factors when investigating the association between dysmagnesemia and mortality? If not, it seems reasonable to mention this as a limitation.

Due to the retrospective nature of the study, it is not possible to precisely investigate which therapies were administered to the patients. However, magnesium supplementation was certainly not given, as hypomagnesemia is often underestimated in hospitalized patients and is not routinely replenished. We mentioned as a limitation of the study, as suggested. Lines 391-417.

  1. The authors noted that the iMg levels at admission and during hospitalization were collected from medical records. Could the authors please clarify if they made a distinction between admission and hospitalization when assessing the outcomes related to iMg?

No, because both patients who presented as hypomagnesemic or hypermagnesemic and those who developed dysmagnesemia during hospitalization were considered dysmagnesemic, as the electrolyte levels were monitored throughout the hospitalization period.

  1. Regarding statistical analysis: Could the authors clarify how the association between dysmagnesemia and outcomes was assessed? For example, would regression analysis be appropriate for assessing the likelihood of mortality in relation to the degree of dysmagnesemia and other outcomes?

Thank you a multiple logistic regression was done to evaluate, we inserted the results of the analisys.Lines 127-147.

  1. Please ensure that the Materials and Methods section of the manuscript clarifies whether a range of renal/urinary diseases (e.g., urethral obstruction, ACKD, AKI, CKD, ESRD) were included under the "urinary" category. It could be confusing for readers if all of these conditions, which impact iMg differently, were grouped into this category.

Thank you for your suggestion. Indeed, all renal/urinary conditions, including urethral obstruction, ACKD, AKI, CKD, and ESRD, were grouped under the 'urinary' category in our analysis. We acknowledge that these conditions may influence ionized magnesium levels differently. This decision was made to simplify the analysis; however, we mentioned this as a limitation of the study in the revised manuscript to avoid any confusion for readers. Lines 391-417. 

Round 2

Reviewer 2 Report

Comments and Suggestions for Authors

Dear authors

Thank you for your updated manuscript and the effort you took in answering my questions and refining the manuscript. I think separating into 2 manuscripts is a great idea.

I still have a few comments before I can recommend the manuscript for publication. 

96: English (nor instead of or)

M&M: I think genereally the M&M is not enough detailled. 

* The sub paragraph is entitled 2.3 although there is no 2.1 nor 2.2…

* I am still missing the inclusion criteria for the enrollment of patient. If you took all patient hospitalized in the ICU please write it clearly in the M&M. At the moment it reads like you took all patients that received a blood gas analysis, which does not imply they were critically ill.

* Please specify what you mean with outcome: death, discharge, survival? 14-day survival?

* What do you mean with SBP and MAP? Which one did you take? Admission? Discharge? Change over 24h?

* What is a "systemic disease"?

Results: 

* 106-107: the p value is also missing…. Was that really statistically significant? What applies to dogs applies to cats as well…

* 111-114: p values are also missing. Please be precise when you talk about statistical results.

* You looked at age, sex, breed, presence of comorbidities… did all of those have an influence on dysmagnesemia? A table could be nice to show the « easy » results.

Author Response

Dear authors

Thank you for your updated manuscript and the effort you took in answering my questions and refining the manuscript. I think separating into 2 manuscripts is a great idea.

I still have a few comments before I can recommend the manuscript for publication. 

96: English (nor instead of or)

Dear reviewer, thank you for you comments. We have corrected the sentence as suggested.

M&M: I think genereally the M&M is not enough detailled. 

* The sub paragraph is entitled 2.3 although there is no 2.1 nor 2.2…

Thank you for your correction. We accidentally missed it during the manuscript review.

* I am still missing the inclusion criteria for the enrollment of patient. If you took all patient hospitalized in the ICU please write it clearly in the M&M. At the moment it reads like you took all patients that received a blood gas analysis, which does not imply they were critically ill.

Yes, we included all patients admitted to the ICU, so we have modified the title of the article to avoid misleading readers and specified in the manuscript that all medical records were analyzed.

* Please specify what you mean with outcome: death, discharge, survival? 14-day survival?

We considered the outcome as whether the patient died, either by euthanasia or naturally, or was discharged from the ICU. We did not evaluate survival beyond 14 days. We have specified this point in the manuscript to avoid any misunderstanding for the readers.

* What do you mean with SBP and MAP? Which one did you take? Admission? Discharge? Change over 24h?

We considered SBP and MAP at the time of ICU admission. Unfortunately, due to the retrospective nature of the study, it was not possible to evaluate the blood pressure trends during hospitalization. As for the previous point, we have specified this in the manuscript.

* What is a "systemic disease"?

By systemic disease, we primarily referred to septic patients, as well as those with SIRS, DIC, multiple organ dysfunction, or other conditions with a systemic impact. For example, trauma patients were included in this category only when presenting with systemic complications beyond musculoskeletal involvement.

Results: 

* 106-107: the p value is also missing…. Was that really statistically significant? What applies to dogs applies to cats as well…

We are sorry but in this sentence  “The mean age was 6.8 ± 4.9 years, and the median weight was 4.5 kg (range 0.3 – 10 kg).” We do not understand where the reviewer ask to point out the p value. This is a descriptive statistic.

* 111-114: p values are also missing. Please be precise when you talk about statistical results.

Thank you for your suggestion, we corrected the sentence.

* You looked at age, sex, breed, presence of comorbidities… did all of those have an influence on dysmagnesemia? A table could be nice to show the « easy » results.

While we recorded data on age, sex, breed, and comorbidities, no significant influence of age or comorbidities on dysmagnesemia was observed in either dogs or cats. Given the lack of any notable impact, these factors were not further analyzed in detail, and thus were not included in the manuscript. Regarding breed, we identified the breeds with the highest incidence of hypomagnesemia in dogs and hypermagnesemia in cats. We believe that including a table would result in redundancy with the information already provided.

Reviewer 3 Report

Comments and Suggestions for Authors

Thank you for incorporating the comments from my previous review. The manuscript has improved significantly, though I still have a few minor comments for the authors' consideration:

  1. The authors mentioned that the established reference range was removed from the manuscript following another reviewer's suggestion. However, the established reference interval (RI) of 0.6-1.11 mmol/L in cats is still included in line 16. Please clarify this point and double-check the manuscript for any other inconsistencies.

  2. In lines 63-65, the authors state that the manufacturer's reference range for both dogs and cats is 0.5-1 mmol/L. I have used the same machine (Stat Profile Prime Plus Critical Care Analyzer, REF 60278), and the manufacturer's instructions for use (IFU) provided a range of 1.09-1.45 mg/dL (0.45-0.59 mmol/L). Please provide supporting evidence for the 0.5-1 mmol/L reference range, either as a reference or in supplementary material, given its importance in defining normal iMg levels and discussing the significance of hyper/hypomagnesemia in the current manuscript.

Author Response

Thank you for incorporating the comments from my previous review. The manuscript has improved significantly, though I still have a few minor comments for the authors' consideration:

  1. The authors mentioned that the established reference range was removed from the manuscript following another reviewer's suggestion. However, the established reference interval (RI) of 0.6-1.11 mmol/L in cats is still included in line 16. Please clarify this point and double-check the manuscript for any other inconsistencies.

Dear reviewer, thank you for your correction. We accidentally missed it during the manuscript review.

  1. In lines 63-65, the authors state that the manufacturer's reference range for both dogs and cats is 0.5-1 mmol/L. I have used the same machine (Stat Profile Prime Plus Critical Care Analyzer, REF 60278), and the manufacturer's instructions for use (IFU) provided a range of 1.09-1.45 mg/dL (0.45-0.59 mmol/L). Please provide supporting evidence for the 0.5-1 mmol/L reference range, either as a reference or in supplementary material, given its importance in defining normal iMg levels and discussing the significance of hyper/hypomagnesemia in the current manuscript.

Dear reviewer, thank you for your observation. We found that the manufacturer’s instructions report an iMg reference range of 0.45-0.6 mmol/L, but the table where this range is listed appears to be referring to human medicine. Furthermore, the manufacturer specifies that the iMg reference range was determined based on data from several institutions that have used Nova analyzers.

However, during our study, we noted that the blood gas analyzer indicated values as "out of range" when iMg levels were below 0.5 mmol/L or above 1 mmol/L. This practical observation led us to adopt a reference range of 0.5-1 mmol/L for our analysis.

We hope this clarifies our approach and provides the necessary context for the reference range used in the study.